# Land reclamation and its consequences: A 40-year analysis of water residence time in Doha Bay, Qatar

Mathieu Lecart[1], Thomas Dobbelaere[1], Lauranne Alaerts[1,2], Ny Riana Randresihaja[1,2], Aboobacker Valliyil Mohammed[3], Ponnumony Vethamony[3], Emmanuel Hanert[1,4]*

**1** Earth and Life Institute (ELI), UCLouvain, Louvain-la-Neuve, Belgium, **2** Department of Astrophysics, Geophysics and Oceanography (AGO), ULiège, Liège, Belgium, **3** Environmental Science Center, Qatar University, Doha, Qatar, **4** Institute of Mechanics, Materials and Civil Engineering (IMMC), UCLouvain, Louvain-la-Neuve, Belgium

* emmanuel.hanert@uclouvain.be

**Data Availability Statement:** The residence time model outputs for each coastal topography and for both summer and winter are available at https://doi.org/10.5281/zenodo.10453435.

## Abstract

Qatar's rapid industrialization, notably in its capital city Doha, has spurred a surge in land reclamation projects, leading to a constriction of the entrance to Doha Bay. By reducing and deflecting the ocean circulation, land reclamation projects have reduced the effective dispersion of wastewater introduced into the bay and hence degraded the water quality. Here, we assess fluctuations in water residence time across three distinct eras (1980, 2000, and 2020) to gauge the impact of successive land reclamation developments. To do this, we couple the multi-scale ocean model SLIM with a Lagrangian model for water residence time within Doha's coastal area. We consider three different topographies of Doha's shoreline to identify which artificial structures contributed the most to increase water residence time. Our findings reveal that the residual ocean circulation in Doha Bay was predominantly impacted by northern developments post-2000. Between 1980 and 2000, the bay's residence time saw a modest rise, of about one day on average. However, this was followed by a substantial surge, of three to six days on average, between 2000 and 2020, which is mostly attributable to The Pearl mega artificial island development. Certain regions of the bay witnessed a tripling of water residence time. Given the ongoing population expansion along the coast, it is anticipated that the growth of artificial structures and coastal reclamation will persist, thereby exacerbating the accumulation of pollutants in the bay. Our findings suggest that artificial offshore structures can exert far-reaching, non-local impacts on water quality, which need to be properly assessed during the planning stages of such developments.

## Introduction

The Arabian/Persian Gulf (hereafter the Gulf, Fig 1a) has undergone a rapid transformation over the past two decades, propelled by the surge in oil and gas extraction. This rapid development has led to extensive coastal alterations, including the construction of artificial islands,

**Funding:** This work was jointly carried out by QU and UCLouvain under the Collaborative Grant project (QUCG-ESC-22 23-591), funded by QU. Computational resources were provided by the Consortium des Equipements de Calcul Intensif (CECI), funded by the FRS-FNRS under Grant No. 2.5020.11. The funders had no role in the study design, data collection and analysis, decision to publish or preparation of the manuscript.

**Competing interests:** The authors declare that they have no known competing financial interests or personal relationships that could have appeared to influence the work reported in this paper.

waterfronts, causeways, and ports [1]. These developments were facilitated through a combination of land reclamation, dredging, and construction activities. However, these modifications have not been without environmental consequences. They have resulted in significant marine habitat loss, severely affecting coral reefs, seagrass meadows, and mangroves [2, 3]. Moreover, these alterations have had far-reaching impacts, as they have modified ocean circulation and wave propagation patterns. Changes in the dynamics of currents and waves have influenced the transport of sediments, biological material, and energy across separated areas within the seascape [4]. These processes have led to an increase in sediment volume upstream of the structures and a decrease downstream, alterations to larval connectivity pathways, and changes in pollutant dispersal patterns [5].

Doha Bay (hereafter the Bay), a semi-enclosed ocean body bordering Qatar's capital city of Doha, epitomizes the rapid transformations that have occurred in the Gulf's coastal topography. The Bay's distinctive crescent-shaped shoreline is a product of a substantial land reclamation project, initiated in 1974, which spanned 750 hectares and led to the establishment of the Diplomatic Quarter, also known as West Bay, at its northern extremity [6] (Fig 1b, location 3). The pace of development within the Bay intensified towards the end of the 1990s, leading to the reclamation of more land along the coastline to develop new projects such as Hamad International Airport and Doha's Port extension to the south (Fig 1c, locations 7 and 8). This was subsequently followed by the major undertakings of Lusail City and The Pearl projects, to the north of the Bay, in the mid-2000s ([7], Fig 1d, locations 9 and 10). The continuous modification of Doha's shoreline topography over the last 40 years has resulted in an increased constriction of the Bay's entrance that reduced the water exchanges with the Gulf.

Ref. [8] followed a stakeholder participatory approach to highlight the detrimental impacts of these significant land reclamation projects. Their findings revealed that land reclamation projects, when executed without adequate planning, can inflict severe damage on coastal and marine environments. Consequences include heightened sedimentation, odor, and turbidity, coupled with pH value fluctuations. These conditions could potentially disrupt the food chain and deplete oxygen levels crucial for marine life survival. Similar effects of artificial structures on the surrounding water quality have been observed elsewhere in the Gulf as well. This is for instance the case around the Palm Jumeirah Island (Dubai, UAE) where the altered current patterns in the surrounding waters have reduced wastewater dispersal, hence leading to nutrient enrichment, increased concentrations of Chlorophyll-a and decreased levels of dissolved oxygen [9, 10]. More broadly, several studies have underscored the detrimental impacts of land reclamation on water quality, particularly when carried out without comprehensive planning. Notable examples include reclamation projects in Qinzhou and Bohai Bay in China [11, 12], and along the Shihwa coast in the Republic of Korea [13].

Water residence time is a simple diagnostic to assess water quality in a particular coastal areas such as a bay, a lagoon or an estuary [14] It measures the time that a water parcel spends in a particular system before leaving it through a defined boundary [15, 16]. In the context of water quality, water residence time is important because it influences the dilution and dispersion of pollutants [17]. If the residence time is short, pollutants that enter the water are quickly flushed out and dispersed into a larger body of water, reducing their concentration and potential impact on water quality. However, if the residence time is long, pollutants can accumulate, leading to higher concentrations that can negatively impact water quality. Semi-enclosed coastal areas like bays or lagoons often receive pollutants from various sources, such as urban runoff and industrial discharges. If these areas also have long water residence times due to factors like limited circulation with the open ocean, these pollutants can build up and lead to significant water quality issues [18]. The accumulation/trapping of particles can also occur due to

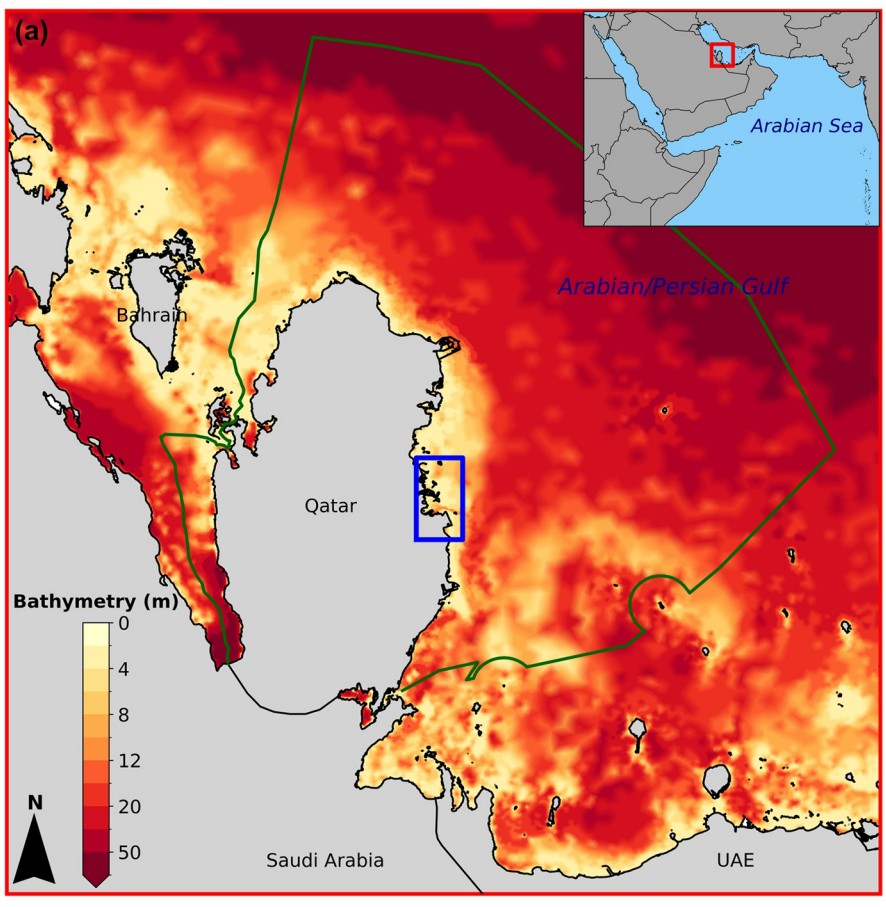

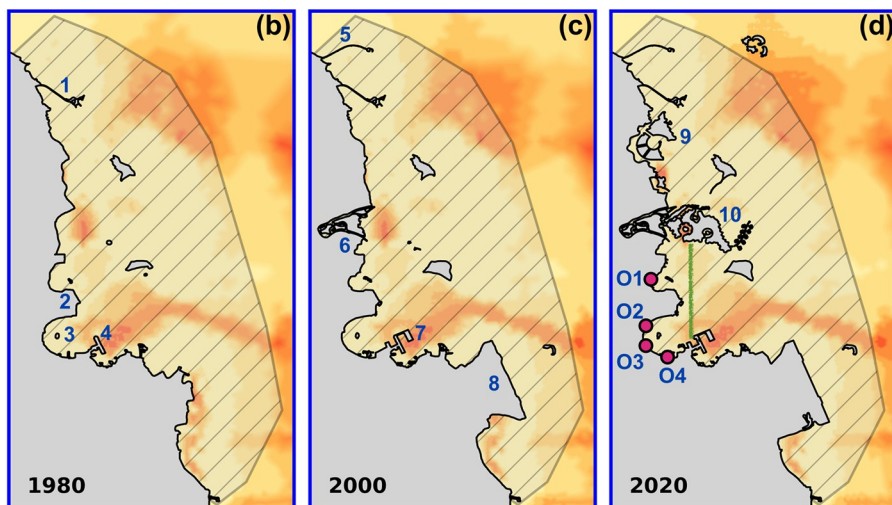

**Fig 1.** (a) The central Gulf's bathymetry, featuring Qatar's Exclusive Economic Zone (EEZ, depicted in green), and detailed views of the coastal area offshore of Doha with the topographies of (b) 1980, (c) 2000 and (d) 2020. Particular locations highlighted in the close-up views are: 1. Lusail causeway #1, 2. West Bay, 3. Doha Bay, 4. Old Doha Port, 5. Lusail causeway #2, 6. Legtaifiya Lagoon, 7. Doha Port extension, 8. Hamad International Airport, 9. Lusail City and 10. The Pearl. The primary wastewater outfalls in Doha, represented by magenta dots in panel (d), are: Diplomatic Area (O1), Tennis Court (O2), Al-Rumaila (O3), and Souq Waqif (O4). The hatched area shown in the detailed views is where water residence time is computed. The green line in panel (d) represents the location of the transect along which residence time values are computed in Fig 5. The base map was sourced from OpenStreetMap and OpenStreetMap Foundation. This figure contains information from OpenStreetMap and OpenStreetMap Foundation, which is made available under the Open Database License (ODbL).

stronger coastward component of eddy circulations, as seen in the case of Gulf of Kachchh, the northwestern Arabian Sea [19].

The objective of this study is to assess how coastal reclamation projects have modified water residence time in the Bay. To achieve this objective, we simulated the ocean circulation in Doha's coastal area for the coastal topographies of 1980, when the Bay had already its crescent shape but prior to all the major development projects, 2000, once the reclamation work for the new airport and port extension had already been initiated and 2020, after the completion of Lusail City and The Pearl projects. We select these three distinct years as representative milestones of Doha Bay's developmental trajectory: the initial situation with limited reclamation, followed by extensive reclamation in the southern part of the Bay, and then a similar expansion in the northern part. The model's spatial resolution reaches less than 50 m along the shoreline and hence accurately represent the impact of artificial offshore structures on the ocean circulation patterns. We subsequently used the simulated currents to calculate water residence time in the Bay and hence estimate its increase due to land reclamation.

## Materials and methods

### Study area

Qatar's peninsula is located in the central part of the Gulf, a semi-enclosed, hypersaline and shallow ocean basin connecting the Indian Ocean through the Strait of Hormuz and Sea of Oman (Fig 1a). The surrounding waters of Qatar are generally not deeper than 20 meters, with the Bay's depth not exceeding 10 meters. This shallow depth makes the Bay particularly conducive to land reclamation projects. The period between 1980 and 2000 saw the first major developments within the Bay, primarily due to land reclamation for the construction of Hamad International Airport and Doha's Port extension [7]. Both of these projects were located in the Bay's southern region (Fig 1c, locations 7 and 8). During this time, the Legtaifiya Lagoon (Fig 1c, location 6) was also completed, although its construction was primarily land-based. By 2020, the development of The Pearl and Lusail City was largely completed (Fig 1d, locations 9 and 10). These projects significantly altered the topography of the Bay's northern region.

In the Gulf, the large-scale circulation is predominantly driven by wind and density gradients [20–22]. However, in shallow coastal regions like those surrounding Qatar, the circulation becomes more influenced by tide and wind. The bathymetry and features of the coastline then become crucial in modifying the circulation, leading to the generation of tidal eddies and local flow intensification exceeding 1 m/s through narrow passages, such as those between breakwaters or within artificial waterways [23]. The Gulf's primary wind patterns are northerly and northwesterly, with the Shamal winds contributing significantly [24–26]. These Shamal winds persist over Qatar during both summer and winter seasons, although they occur more frequently and with greater intensity during the winter [27, 28]. Apart from the Shamal winds, the easterly Nashi winds have significant influence along the east coast of Qatar [29].

In addition to being vulnerable to oil spills from the Gulf [30], Doha Bay is subject to pollutant influx from a variety of local sources, including wastewater discharges from sewage and water treatment facilities. Four primary outfalls—*Souq Waquif, Al-Rumaila, Tennis Court,* and *Diplomatic Area*—serve as conduits for municipal sewage and stormwater, thereby continuously introducing pollutants into the Bay (Fig 1d). A water quality risk assessment conducted by [31] underscored the eutrophication risk near the Souq Waqif outfall, which drains nutrients from a considerably larger area compared to the other three outfalls. The study also emphasized the environmental risk tied to the accumulation of particle-bound heavy metals and other hazardous pollutants within the harbour area at the Souq Waqif outfall and within 250 m of the Al-Rumaila, Tennis Court, and Diplomatic outfalls.

## Ocean circulation model

In this study, we employ the multi-scale ocean model SLIM (Second-generation Louvain-la-Neuve Ice-ocean Model, www.slim-ocean.be) to simulate the ocean circulation around Qatar. SLIM solves the ocean circulation governing equations on an unstructured mesh, which allows for locally increased resolution to accurately capture variations in bathymetry and coastal topography [32, 33]. Our focus is on simulating the ocean circulation within Doha's coastal area, but our model encompasses a wider area, extending approximately 200 km east of Qatar. Given the area's shallow depth—less than 10 m—we utilize the 2D barotropic version of SLIM [34, 35]. The model's mesh achieves a horizontal resolution of about 50 m along Doha's shoreline and around artificial structures (Fig 2). As the complexity of the coastline topography increases with the addition of new artificial structures, the number of mesh elements correspondingly increases. Specifically, the meshes for the 1980, 2000, and 2020 topographies comprise approximately 2.7e4, 3.4e4, and 4.5e4 elements, respectively.

In our model, we parameterize the bottom stress using a Chézy-Manning drag formulation with a Manning coefficient of $n = 0.025$ m$^{-1/3}$s. The model is driven by wind data sourced from the European Centre for Medium-Range Weather Forecasts' (ECMWF) ERA5 reanalysis, which offers a spatial resolution of 31 km and a temporal resolution of 1 hour. At the model's open boundary, we impose a combination of depth-averaged velocity and sea surface elevation

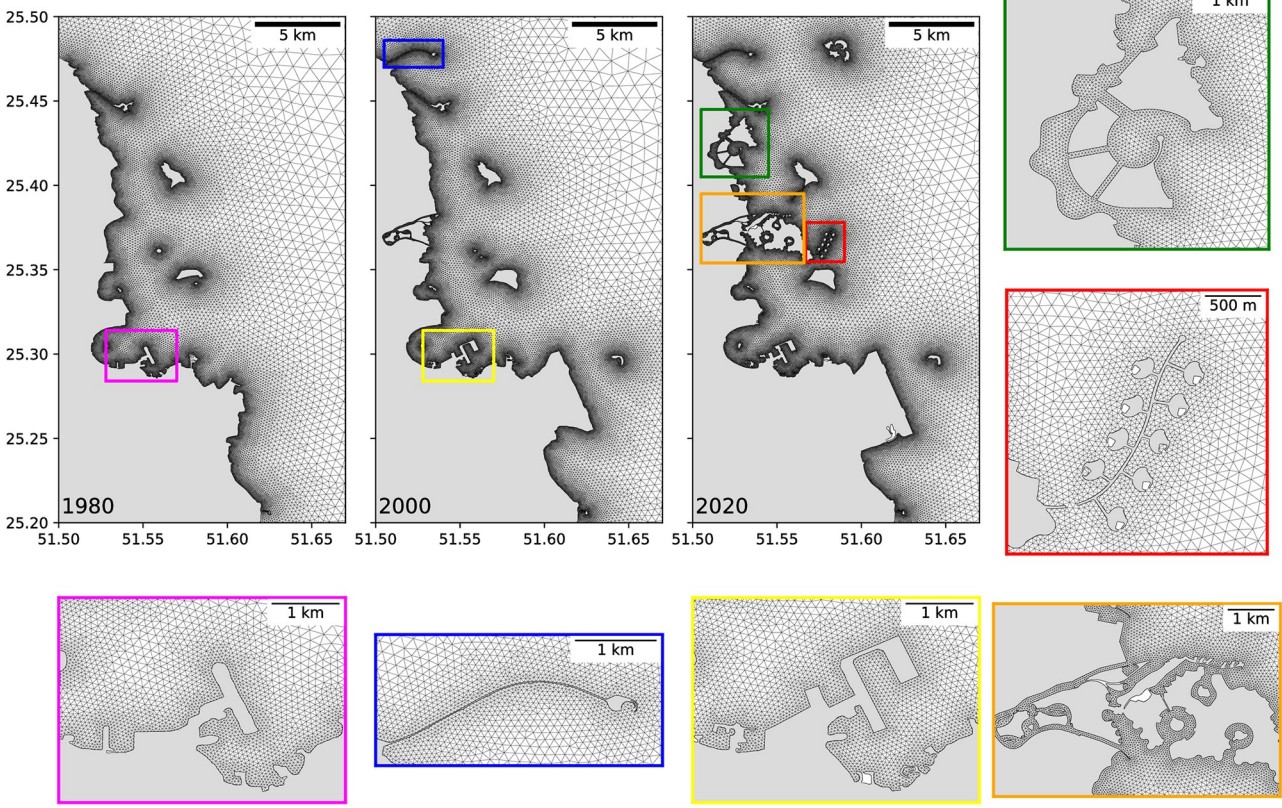

**Fig 2. Model mesh in Doha's coastal area, incl.** The Bay, for the coastal topographies of 1980, 2000 and 2020, with close-up views on the main artificial structures. The mesh resolution is about 50 m along the shoreline and around artificial structures. The base map was sourced from OpenStreetMap and OpenStreetMap Foundation. This figure contains information from OpenStreetMap and OpenStreetMap Foundation, which is made available under the Open Database License (ODbL).

provided by the Mercator global ocean analysis, which is generated by running the global data-assimilated ocean model NEMO on a 1/12˚ grid, and from the OSU TPXO9-atlas tidal dataset [36]. The model's bathymetry is extracted from the GEBCO 2022 global terrain model, which offers a resolution of 15 arc-seconds [37]. For the purposes of this study, we considered the bathymetry dataset to be representative of the conditions in 2020. Given the lack of available bathymetry data for 1980 and 2000, we made the assumption that the bathymetry for these years mirrored that of 2020. Furthermore, we posited that areas where artificial structures had not yet been established would have a depth of 2 meters. This is obviously an approximation but that depth is representative of water depth along Doha's shoreline. It is further common for land reclamation projects to take place in relatively shallow waters to minimize the amount of fill needed and to control costs. Our model has been validated against sea surface elevation and current velocity observations in Qatar's coastal waters [23]. For additional validation in Doha Bay during the time period considered in this study, see S1 Fig. For additional details on the model formulation, see [23].

## Water residence time model

Residence time is typically defined as the duration required for a parcel of water to exit a specified region of interest for the first time [38]. Given that parcels originating from different locations and times within the region may take varying durations to exit, residence time is inherently a function of both location and time. It can be calculated using either a forward-in-time or backward-in-time numerical integration [39]. In this study, we employ a forward Lagrangian particle-tracking approach. This method involves releasing a large number of virtual particles in each mesh element within the area of interest and tracking them until they exit the area. The residence time of each particle is then recorded as the time elapsed between its release and its departure from the area of interest (see extent in Fig 1). This approach has been utilized in several studies to estimate water residence time in coastal systems, *e.g.* [40–42].

Our Lagrangian particle tracker employs two-dimensional ocean currents to advect virtual particles from their release points, incorporating a diffusive component expressed as a random walk with a diffusivity of $K = \alpha \Delta^{1.15}$, where $\alpha = 2 \times 10^{-4}$ m$^{0.85}$/s and $\Delta$ represents the local mesh resolution [43]. As this represents a depth-averaged transport model, the transport velocity must include bathymetry and diffusivity gradient components to be consistent with the 2D Eulerian advection-diffusion equation [44]. Over a 15-day period, we released one particle every three hours across all mesh elements within our area of interest. This extended release period ensured that the particles experienced a wide spectrum of tidal conditions throughout the entire spring-neap cycle. Consequently, a total of 120 particles were released across all mesh elements within the area of interest, which yields a total of about $4.36 \times 10^5$, $5.39 \times 10^5$ and $6.49 \times 10^5$ particles for the 1980, 2000 and 2020 meshes, respectively. We then simulated the transport of these particles over a three-month period. By the end of the simulation, nearly all particles had exited the area of interest.

For each particle released from a mesh element with index $i$, we computed the time it takes for the particle to exit the area of interest. At the end of the simulation, we computed the average of these durations to derive a unique value of the residence time, denoted as $\theta_i$, for each mesh element within the area of interest. The residence time distribution is therefore a piecewise constant function. Given the fine mesh resolution in the area of interest, this piecewise constant representation of the residence time produces smooth results. Once we have the residence time distribution, we can calculate its average value over the entire area of interest (or a

portion thereof) using the formula:

$$\bar{\theta} = \frac{\sum_{i=1}^{N} \theta_i A_i}{\sum_{i=1}^{N} A_i},$$

where $N$ represents the number of mesh elements in the area where the average is computed, and $A_i$ denotes the surface area of element $i$.

## Results

We assessed the changes in water residence time between 1980, 2000 and 2020 by simulating the ocean circulation in the coastal area surrounding Doha Bay for the topographies of those three years. The ocean circulation in this area is predominantly tidally-driven, yet it is also influenced by wind patterns. These wind patterns exhibit seasonal variations, largely governed by the Shamal winds, which intensify during winter and subside in summer [27]. We hence simulated ocean currents under both winter (01 December 2021—28 February 2022) and summer (01 June—31 August 2022) wind conditions. These forcings were then maintained across all three topographies to isolate the influence of topographical changes on the ocean circulation and the subsequent impact on the residence time distribution.

### Ocean circulation patterns

Alterations to the Bay's circulation patterns, driven by topographical modifications, can be visualized through the computation of residual flow patterns. These patterns were derived by averaging the simulated ocean currents over the three-month simulation period for both winter and summer seasons (streamlines in Fig 3). By doing that, we obtain a picture of the net movement of water over each season and hence better identify how currents are moving water masses through the area. However, since the tidal currents are the most energetic near the coast, we also computed the mean current speed obtained by averaging the currents amplitude over each season (background color in Fig 3). In that case, the effect of the tides is not filtered out and we can identify areas where the flow is most intense.

The offshore residual flow patterns are mostly southward. However, on approaching the shore, these flows interact with the coastal topography and are deflected by artificial structures. For instance, Lusail's causeway #1 (already present in 1980) and #2 (present from 2000) deflect the southward flow eastward, but only over a limited distance. In the scenarios of 1980 and 2000, the flow resumes its southward direction south of Lusail's causeways, extending all the way to the Bay, where it veers east, forming a "leaky" eddy just north of Doha's Port. Streamlines can escape this eddy and merge with the offshore southward flow. However, the flow patterns in 2020 significantly differ as The Pearl development acts as a barrier to the southward flow, preventing it from reaching the Bay. Instead, it is deflected eastward, where it merges with the offshore circulation. South of The Pearl, the circulation is weaker and consists of smaller eddies. These eddies also appear to be less "leaky" than the one present in 1980 and 2000. It's also noteworthy to mention the intensification of the flow in Legtaifiya Lagoon and between the main island of The Pearl and the cluster of smaller islands to the east.

Upon comparing these various flow patterns, it appears that the differences are more pronounced for different topographies than different seasons with the same topography. This observed limited seasonal variability supports the assertion that the circulation close to the shore is primarily driven by tidal forces. The residual flow, which is directed southward, remains mostly unaffected by the southern developments in the Bay that occurred between 1980 and 2020, such as the significant land reclamation for the airport construction. Consequently, alterations to the Bay's flow patterns between 1980 and 2000 are relatively minor.

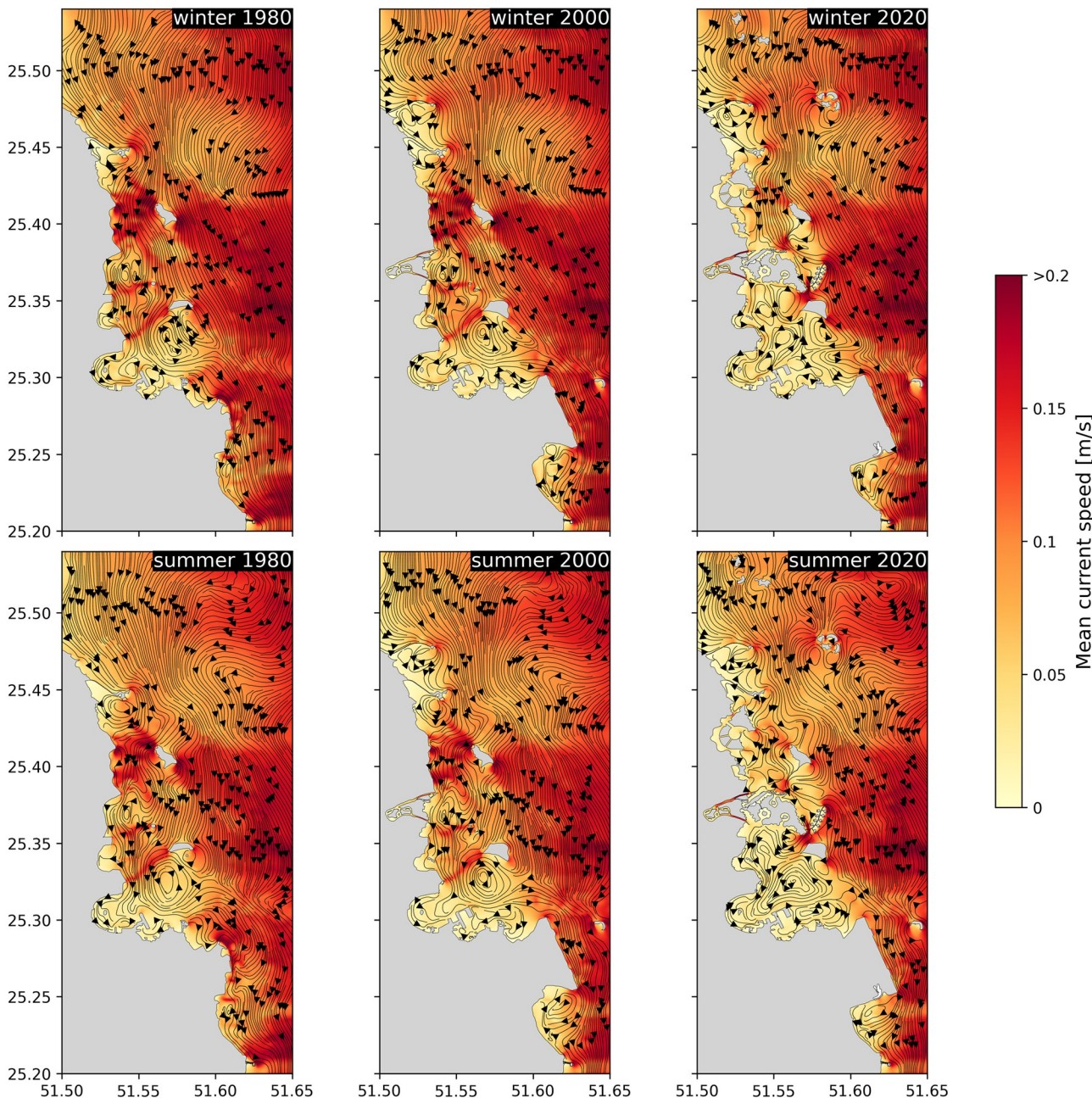

**Fig 3. Residual flow circulation patterns (black streamlines) and mean current speed in the Bay for winter (top) and summer (bottom), and for the topographies of 1980, 2000 and 2020.** The colorbar is the same for all figures. The base map was sourced from OpenStreetMap and OpenStreetMap Foundation. This figure contains information from OpenStreetMap and OpenStreetMap Foundation, which is made available under the Open Database License (ODbL).

However, from 2000 to 2020, most reclamation projects were situated north of the Bay, thereby exerting a greater impact on the incoming flow to the Bay. The cumulative effect of these changes results in a sheltering of the Bay from the southward flow and a weakening of the overall circulation within the Bay.

## Residence time patterns

Alterations to ocean circulation patterns directly influence the residence time. Areas that are sheltered from the main circulation by artificial structures have a larger residence time. For instance, the region immediately south of Lusail's causeway #1 (Fig 1b, location 1) had a residence time exceeding 30 days already in 1980 (Fig 4). In contrast, the remaining coastal area had a relatively short and uniform residence time in 1980, with a mean of 5–7 days and a standard deviation of less than 5 days. Overall, the residence time is a bit longer in winter than in summer. Water from the Bay required approximately 8–10 days to exit the area of interest.

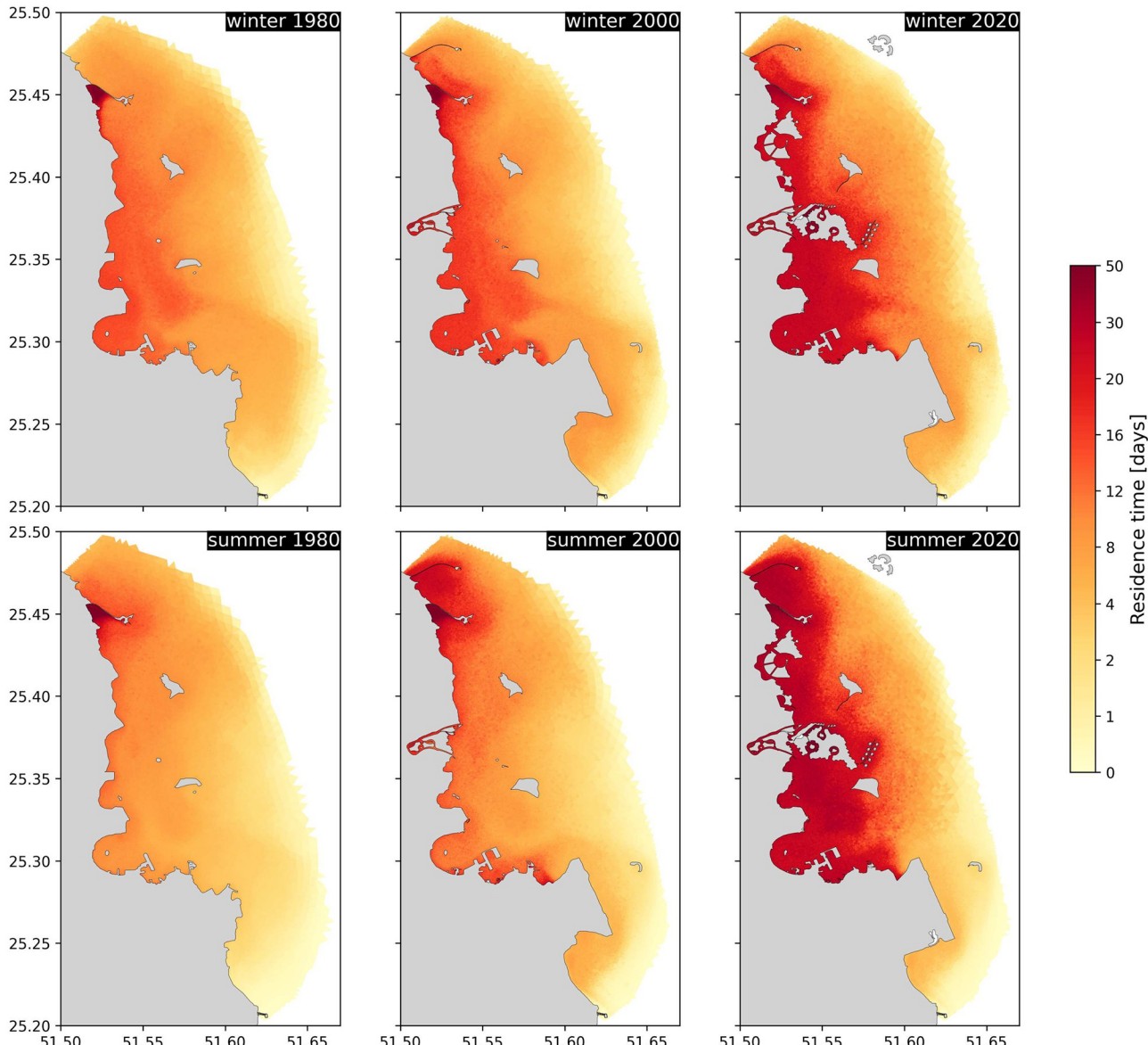

**Fig 4. Residence time distribution in the Bay for winter (top) and summer (bottom), and for the topographies of 1980, 2000, 2020.** The colorbar is the same for all figures. The base map was sourced from OpenStreetMap and OpenStreetMap Foundation. This figure contains information from OpenStreetMap and OpenStreetMap Foundation, which is made available under the Open Database License (ODbL).

Between 1980 and 2000, developments were primarily concentrated in the southern part of the Bay, thus having a minimal impact on the southward residual circulation entering the Bay. As a result, the distribution of residence time in Doha's coastal area experienced minor alterations during this period (Fig 4). On average, it saw an increase of roughly one day (+10% in winter and +25% in summer). These increases were chiefly concentrated in areas immediately adjacent to the new structures, such as the regions sheltered by Lusail's causeway #2, the port extension, and south of the airport reclamation (Fig 1c, locations 5, 7 and 8). Consequently, the standard deviation rose by 16% in winter and 25% in summer. Despite a more substantial relative increase in summer, the patterns of residence time remained fairly consistent across both seasons.

From 2000 to 2020, the residence time experienced a more marked increase. On average, across the entire area of interest, it rose by approximately 3 days (+40%) in winter and 6 days (+79%) in summer. However, these average changes conceal significantly larger local differences. The creation of the Lusail City artificial islands (Fig 1c, location 9) reduced the circulation along the coast and just east of these islands, leading to an increase in the residence time in that area. There is now an approximately 2 km wide swath where the residence time exceeds 20 days along the entirety of Lusail's shoreline. The most substantial increase, however, pertains to the entire coastal area located south of The Pearl development (Fig 1c, location 10), which encompasses the Bay. This area undergoes a substantial increase, with residence times soaring from about 10 days to over 25 days in an area covering roughly 25 km$^2$ between The Pearl and Doha's Port at the southern end of the Bay. This increase is clearly evident when calculating the residence time along a north-south transect from The Pearl to Doha's Port (Fig 5, see transect location in Fig 1d). In winter, the residence time rose from about 13 days in 1980 to approximately 24–25 days in 2020 (an 80–90% increase). In summer, it escalated from about 8–9 days to roughly 25–30 days (a 200–250% increase). Interestingly, the increase is not confined to the immediate vicinity of the structure but extends over more than 7 km away. More locally, the water residence time in some of the blind channels and pathways within The Pearl, such as the main marina, approaches nearly 50 days.

To evaluate more accurately the potential water quality concerns near the wastewater outfalls in the Bay, we also calculated the average residence time within a 1-km radius of each outfall (Fig 6). The average residence times across the four outfalls are quite similar. The difference between the conditions in 1980 and those in 2000 is also relatively minor. For both winter and summer conditions, we note an increase of approximately 2 to 3 days. Although the absolute changes are comparable, the average residence time in winter is about 5 days longer than in summer. When comparing these figures with those obtained using the 2020 topography, we once again notice more substantial changes, with an increase of roughly 8 days in winter and 12 days in summer. This results in average residence times exceeding 25 days for all four outfalls.

## Discussion and conclusions

Over the last 40 years, land reclamation initiatives along Doha's coastline have resulted in an increased water residence time within Doha's coastal waters and in the Bay in particular. This increase is primarily attributed to projects undertaken after 2000, north of the Bay, namely The Pearl and Lusail City. Earlier developments to the south of the Bay, including the port extension and airport reclamation, had a less significant impact. The alterations in residence time are a direct consequence of changes to the ocean circulation along the coast and within the Bay itself. The artificial islands and offshore developments deflect the southward residual circulation off Doha. Developments to the north of the Bay lead to increased sheltering from

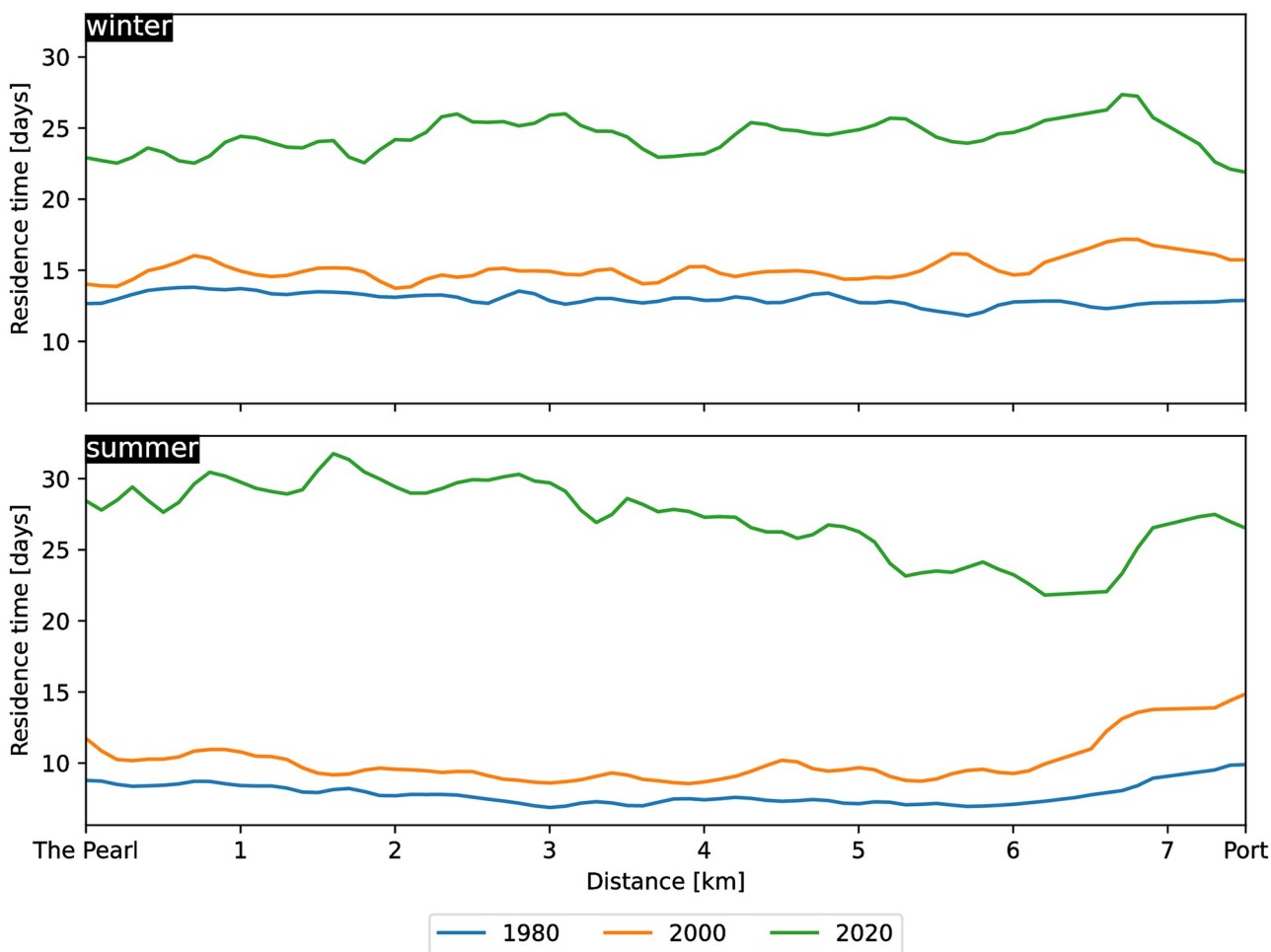

**Fig 5. Residence time values along a 7.5km north-south transect connecting The Pearl to Doha's Port.** It highlights the residence time increase in the entire Bay area following the construction of The Pearl. The location of the transect is shown in Fig 1.

the main circulation, thereby reducing the Bay's flushing capacity. While the Bay's water quality is currently within the acceptable limits [31], our findings indicate a potential for rapid deterioration in the face of Doha's growing population, driven mostly by temporary visitors, coastal-based industries and activities, and future land reclamation projects that could further limit water exchanges with the Gulf.

Our research underscores the extensive impact that inadequately planned artificial structures can have on water quality. The Pearl development, for instance, has led to the sheltering of the Bay from the primary southward ocean currents, resulting in a weakened flushing of the Bay's waters and promoting the accumulation of pollutants discharged into the Bay. From 1980 to 2020, the average rise in water residence time across the studied area, which includes and extends beyond the Bay, spans roughly 4 to 7 days. However, our findings reveal that the water residence time in the region directly south of The Pearl, which encompasses the Bay and Doha's most iconic waterfront, experienced a staggering increase of up to 20 days, effectively more than tripling the initial duration. This surge is primarily attributable to the construction of The Pearl. Notably, this increase is not confined to narrow, blind channels within the artificial structure itself, but extends several kilometers away from the structure.

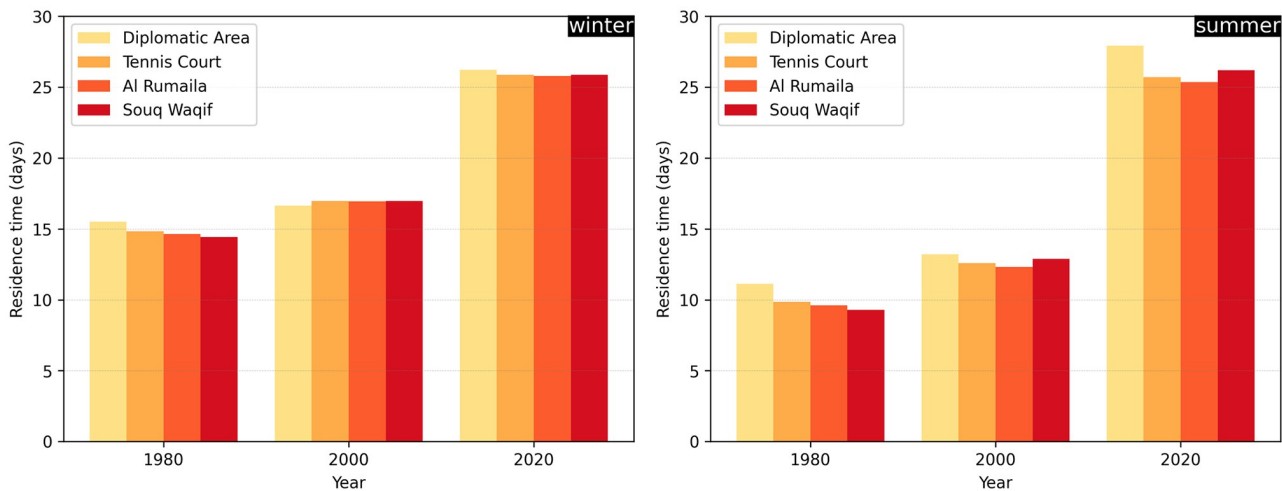

**Fig 6. Average residence time within 1km of each outfall in the Bay for winter (left) and summer (right), and for the coastal topographies of 1980, 2000 and 2020.**

The Pearl, a mega artificial island with a striking geometric design visible from space, exemplifies the wave of such projects that have swept across the Gulf over the past two decades. These projects, designed to capture public attention and offer prime real estate, include The Palm and The World Islands in Dubai (UAE), Durrat Al Bahrain and Diyar Al Muharraq in Bahrain, and Towers Island in Kuwait [45]. During their design phase, the extensive environmental impact of these projects was occasionally underestimated. Beyond introducing new sources of pollution offshore, the construction of these structures led to significant seafloor alterations and the destruction of marine ecosystems, including coral reefs, seagrass meadows, and mangroves [2, 46]. By altering ocean currents, these developments also have non-local consequences. They can exert far-reaching downstream effects, impacting water quality several kilometers away [9]. These observations concur with the findings of Zhang et al. [12] who showed that land reclamations in Bohai Bay (China) significantly altered the residual circulation and tidal prisms. Such alterations led to a decline in water quality, characterized by heightened retention of dissolved inorganic nitrogen in the bay and a rise in phytoplankton carbon concentrations.

To mitigate the environmental impact of land reclamation projects in Gulf countries, a multifaceted approach that balances development with environmental preservation is essential. Enhancing the effectiveness of Environmental Impact Assessment (EIA) is a critical step, where EIA must be comprehensively integrated into decision-making processes, ensuring thorough consideration of environmental impacts and alternatives [47]. Gulf countries should strengthen EIA legislation, including detailed procedural guidelines for assessments, public participation, and monitoring of ecological impacts. This can be achieved by adopting a strategic approach to environmental assessment, which accounts for cumulative effects of multiple projects and involves stakeholders in a more meaningful way, thereby improving water quality management and preserving key ecosystems. Furthermore, establishing a legal framework mandating spatial mapping of sensitive coastal and marine environments would aid in identifying and protecting vulnerable areas from the adverse effects of reclamation activities. By doing so, Gulf countries could ensure a more sustainable approach to development, where the

preservation of biodiversity and ecological integrity is given equal importance as economic growth.

Improving the design of artificial structures is another avenue to reduce their environmental impact. Artificial structures built on land reclamation should include open channels allowing ocean currents to partially flow through the structure. This permeability would facilitate the flushing of the area downstream and hence improve its water quality [48]. Very Large Floating Structures (VLFS) also present an interesting alternative to traditional land reclamation methods, especially in terms of reducing their impact on ocean currents dynamics and subsequently on water quality [49]. They consist in expansive, man-made platforms that float on the water's surface and have been designed for various purposed such as airports, hotels, or agricultural space [50]. While they are most cost-effective in areas where water depth is significant and might thus not be suitable everywhere in the Gulf, their environmental impact is much lower than land reclamation. VLFS do not create physical barriers that could alter ocean currents, which is crucial for maintaining water quality. Furthermore, they do not require the installation of permanent structures over the sea bed, thus preserving benthic habitats. They can further be dismantled if the sea area is needed for other purposes in the future.

As with any modelling study, it is important to acknowledge the limitations and assumptions inherent to our approach. Firstly, our model focuses solely on ocean currents, excluding wind-generated waves. Consequently, we do not account for transport pattern alterations resulting from wave deflection by artificial structures. The impact of waves on transport processes, particularly through the Stokes drift, can be significant for sediment erosion and accretion near structures built perpendicular to the shoreline [51]. The influence of Stokes drift on pollutant transport dynamics in Doha Bay is however expected to be limited. On the one hand, the narrow and curved configuration of the Strait of Hormuz prevents swell waves from penetrating into the Gulf, leading to a wave climate in both the Gulf and Doha Bay that is predominantly shaped by locally generated wind waves [52]. On the second hand, the bay is sheltered from the prevailing southeastward Shamal winds and wave growth is thus fetch-limited as the winds move off the land and back onto the water [53]. Another assumption is that our model employs a two-dimensional barotropic approach, meaning that the vertical ocean circulation dynamics are neglected. This assumption is supported by Doha Bay's shallow depth and significant tidal amplitude. Lastly, we use residence time as a surrogate for water quality. This is an approximation, as water quality is affected by various pollutants, each undergoing distinct chemical reactions like hydrolysis, oxidation-reduction, photolysis, biotic transformation, adsorption, among others [54]. In this study, we implicitly assume that these complex chemical processes occur more slowly than the Bay's flushing by ocean currents. Thus, in the short term, flushing is presumed to be the primary determinant of water quality in the Bay.

While Qatar's population growth is not projected to be particularly strong in the near future [55], Doha is anticipated to continue being a hub for major international events, such as the 2022 World Cup. There are for instance talks of a potential bid to host the 2036 Olympic Games [56]. These high-profile events draw a considerable number of visitors to Doha, thereby increasing the strain on the city's sewage system and the wastewater discharge into the Bay. Simultaneously, other offshore projects are under consideration, which could further alter the Bay's topography. One such project is the Sharq Crossing, a proposed infrastructure development that aims to link West Bay with Hamad International Airport via a 10-kilometer network of bridges and tunnels crossing the Bay [57]. The combination of increased wastewater discharge and the additional narrowing of the Bay's entrance due to these developments could potentially exacerbate water quality issues in the Bay. Therefore, we strongly advise that any future developments along Doha's shoreline should be planned with a comprehensive assessment of their overall environmental impact.

## Supporting information

**S1 Fig. Ocean model validation.** Although the model we employ has previously been validated for sea surface elevation and current velocity within Doha Bay and northeast of Qatar, we had the opportunity to access additional sea surface elevation measurements from the center of the Bay (25˚19'48.0"N, 51˚33'36.0"E) taken between December 8, 2021, and January 17, 2022. This period aligns with the winter season simulated in our model. While the model's generated sea surface elevation exhibits a slightly smaller magnitude than the observed data, the simulation closely mirrors the actual observations. The root mean square error (RMSE) between the simulated and observed elevations amounts to 9.6 cm. This result also indicates that the tidal amplitude within the Bay exceeds one meter. Given the Bay's limited water depth, this suggests that the flow within the Bay is predominantly influenced by the tides. (TIF)

## Author Contributions

**Conceptualization:** Aboobacker Valliyil Mohammed, Ponnumony Vethamony, Emmanuel Hanert.

**Investigation:** Mathieu Lecart, Thomas Dobbelaere.

**Methodology:** Thomas Dobbelaere, Aboobacker Valliyil Mohammed, Emmanuel Hanert.

**Software:** Mathieu Lecart, Thomas Dobbelaere, Lauranne Alaerts, Ny Riana Randresihaja.

**Supervision:** Emmanuel Hanert.

**Writing – original draft:** Emmanuel Hanert.

**Writing – review & editing:** Thomas Dobbelaere, Lauranne Alaerts, Ny Riana Randresihaja, Aboobacker Valliyil Mohammed, Ponnumony Vethamony.

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
