## [Decision Letter · Decision Letter 0]

30 Oct 2023

PONE-D-23-29608Land reclamation and its consequences: A 40-year analysis of water residence time in Doha Bay, QatarPLOS ONE

Dear Dr. Hanert,

Thank you for submitting your manuscript to PLOS ONE. After careful consideration, we feel that it has merit but does not fully meet PLOS ONE’s publication criteria as it currently stands. Therefore, we invite you to submit a revised version of the manuscript that addresses the points raised during the review process.

We look forward to receiving your revised manuscript.

Kind regards,

Tariq Umar, PhD, CEng, EUR ING, MICE, FHEA

Academic Editor

PLOS ONE

[This work was jointly carried out by QU and UCLouvain under the Collaborative Grant 337

project ((QUCG-ESC-22 23-591), funded by QU. Computational resources were 338

provided by the Consortium des Equipements de Calcul Intensif ( ´ ceci ´ ), funded by the 339

f.r.s.-fnrs under Grant No. 2.5020.11.]

 [This work was jointly carried out by QU and UCLouvain under the Collaborative Grant project (QUCG-ESC-22 23-591), funded by QU. Computational resources were provided by the Consortium des Equipements de Calcul Intensif (CECI), funded by the FRS-FNRS under Grant No. 2.5020.11. The funders had no role in the study design, data collection and analysis, decision to publish or preparation of the manuscript.]

6. We note that Figure(s) 2, 3 and 4 in your submission contain copyrighted images. All PLOS content is published under the Creative Commons Attribution License (CC BY 4.0), which means that the manuscript, images, and Supporting Information files will be freely available online, and any third party is permitted to access, download, copy, distribute, and use these materials in any way, even commercially, with proper attribution. For more information, see our copyright guidelines: http://journals.plos.org/plosone/s/licenses-and-copyright.

a. You may seek permission from the original copyright holder of Figure(s) 2, 3 and 4 to publish the content specifically under the CC BY 4.0 license. 

7. We notice that your supplementary figures are uploaded with the file type 'Figure'. Please amend the file type to 'Supporting Information'. Please ensure that each Supporting Information file has a legend listed in the manuscript after the references list.

Additional Editor Comments:

Dear Authors,

We have now received two reviewers comments and recommendations for your paper. I would like to invite to address their comments and submit the revised version of your paper with a detailed response to their comments and how you have addressed them in the revised version.

Tariq

Reviewers' comments:

Reviewer's Responses to Questions

**Comments to the Author**

1. Is the manuscript technically sound, and do the data support the conclusions?

Reviewer #1: Yes

Reviewer #2: Yes

2. Has the statistical analysis been performed appropriately and rigorously? 

Reviewer #1: Yes

Reviewer #2: I Don't Know

3. Have the authors made all data underlying the findings in their manuscript fully available?

Reviewer #1: Yes

Reviewer #2: Yes

4. Is the manuscript presented in an intelligible fashion and written in standard English?

Reviewer #1: Yes

Reviewer #2: Yes

5. Review Comments to the Author

Reviewer #1: The research is related to "Land reclamation in Doha Bay, Qatar and its consequences by conducting water residence analysis," a unique and possesses the potential to add a valuable understanding of the problem specific to Doha Bay, Qatar. However, here are some comments for the authors. They should consider these suggestions to enhance the quality and impact of their research.

1. Introduction:

The section is generally well-written and provides a clear introduction to coastal alterations in the Gulf due to rapid development.

a. Figure1:

• The review suggests adding numbers or indications to the locations mentioned in Figure 1, which would help readers identify them easily.

• There's a recommendation to replace the word "red" with "pink" in the description of the wastewater outfalls to match the figure's color.

b. Adding references:

The review suggests incorporating in-text references to support the claims made in the introduction. For example, adding a reference to "Water residence time is a simple diagnostic to assess water quality in a particular coastal area such as a bay, a lagoon or an estuary".

c. Clarify how the study will transition between the different periods (1980, 2000, and 2020) and the significance of these specific years in coastal development.

2. Results

Ocean circulation patterns

a. The authors should consider how this section connects with other parts of the paper, particularly the sections discussing water residence time and environmental impact. Ensure a smooth transition and integration of key findings and implications.

b. If specific studies or literature influenced the analysis of circulation patterns, integrate citations in the text to support the findings.

Residence time patterns

a. Consider providing more detail on the reasons for the seasonal differences in residence time. Elaborating on these variations and offering potential explanations would strengthen the analysis.

b. Mention if there are other studies or locations with similar findings to provide context and support the significance of the results. Are other regions facing similar challenges with artificial structures and their impact on residence time?

c. Conclude the section by summarizing the key findings and their implications. How do these residence time patterns relate to the overall research question about the impact of land reclamation in Doha Bay?

3. Discussion and conclusions

a. The paper is encouraged to discuss potential mitigation measures. Highlighting recommendations for mitigating the environmental impact of development projects can enhance the paper's practical utility.

b. Consider exploring potential solutions and strategies to address the issues raised in the paper. Offering guidance on balancing development with environmental preservation would be valuable.

c. Suggest specific policy and regulation recommendations based on the research findings. How can environmental impact assessments be improved, or what regulations could mitigate the negative effects of land reclamation?

d. Comparing the findings with similar studies in other regions, especially those with similar development projects, can provide context and make the results more generalizable.

e. The authors must provide near-future recommendations given their results.

Reviewer #2: Abstract: the section presents a good synopsis of the work but could have also use extra sentences to summarize the main results found – although some of these were mentioned in the author’s summary.

Introduction: this sets the scene smoothly well, but the justification of the research could have been more robust - perhaps adding a literature review section.

Methodology: A systematic research method has been used or described. What is not clear is why is this method the best to achieve the aim? Can these lead to generalisable conclusions? 

Results/Discussion/Conclusion:

Good work but the result should be able to show clearly what is new? what is surprising? What was learn when compared to the existing body of knowledge?

What are the limitations and novelty of this paper?

Discussion and conclusion - perhaps should be separated and conclusion should be rewritten to show whether the aim was achieved and identify the contribution to knowledge.

6. PLOS authors have the option to publish the peer review history of their article (what does this mean?). If published, this will include your full peer review and any attached files.

Reviewer #1: No

Reviewer #2: No

---

## [Author Response · Author response to Decision Letter 0]

14 Dec 2023

please see our detailed response to both reviewers' comments attached to our submission as a PDF file.

---

## [Editor Report · Decision Letter 1]

18 Dec 2023

Land reclamation and its consequences: A 40-year analysis of water residence time in Doha Bay, Qatar

PONE-D-23-29608R1

Dear Dr. Hanert,

We’re pleased to inform you that your manuscript has been judged scientifically suitable for publication and will be formally accepted for publication once it meets all outstanding technical requirements.

Kind regards,

Tariq Umar, PhD, CEng, EUR ING, MICE, FHEA

Academic Editor

PLOS ONE
---

## [Editor Report · Acceptance letter]

9 Jan 2024

PONE-D-23-29608R1 

PLOS ONE

Dear Dr. Hanert, 

I'm pleased to inform you that your manuscript has been deemed suitable for publication in PLOS ONE. Congratulations! Your manuscript is now being handed over to our production team.

Kind regards, 

on behalf of

Dr Tariq Umar 

Academic Editor

PLOS ONE